# Probing Clinical Concepts in an EHR Foundation Model via Sparse Autoencoders

**Shashank Yadav** [1]   **David M. Routman** [1]   **Andrew Foong** [1]

## Abstract

Foundation models (FMs) trained on large electronic health record (EHR) datasets can predict patient outcomes, but it is difficult to know what medical knowledge they have acquired. Unlike chatbot LLMs, EHR-FMs are being considered for high-stakes clinical deployment, making it especially important to audit what they have learned beyond predictive accuracy. We apply sparse autoencoders (SAEs) to a transformer-based FM trained on the MIMIC-IV dataset, extending SAE-based mechanistic interpretability to FMs trained on clinical event streams. We use LLM-based interpretation to characterize learned features, suggesting that EHR-FMs organize clinical concepts along an axis distinct from the International Classification of Diseases (ICD) system. We show that learned features are organized by prevalence and that the model encodes candidate matches to known clinical syndromes as single monosemantic features. Syndromic features are composed from lower-level features through cross-layer information-flow circuits that we probe via activation patching. We validate the learned features along two axes: external validity, where feature activations align with held-out ICD phenotypes, and interventional consistency, where activation patching produces measurable downstream effects in source-target pairs. Together, these results demonstrate the utility of SAEs as an interpretive layer for EHR foundation models.

## 1. Introduction

Foundation models (FMs) for electronic health records (EHRs), collectively referred to as EHR-FMs, have been recently developed to learn from diverse clinical data (Stein-berg et al., 2021; Rasmy et al., 2021; Fallahpour et al., 2024; Steinberg et al., 2023). EHR-FMs transfer well to mortality, readmission and phenotyping tasks (Wornow et al., 2023; Steinberg et al., 2023), but it is difficult to inspect what clinical knowledge they encode, whether they capture pathophysiology or spurious correlations, and how they organize medical concepts internally.

Sparse autoencoders (SAEs) offer a principled approach towards interpretability of the learned representations. SAEs decompose neural representations into monosemantic features (Templeton, 2024; Gao et al., 2025). SAEs build on the superposition hypothesis (Elhage et al., 2022) that neural networks represent more concepts than they have dimensions by encoding features in overlapping, superposed patterns that SAEs can disentangle. However, applying SAEs to domain-specific models introduces a challenge that LLM interpretability does not face: *feature interpretation requires domain expertise*.

For EHR-FMs, a feature's top-activating inputs are sequences of structured medical codes rather than natural language. For example, LAB//50912 = 3.2 mg/dL encodes a serum creatinine (a kidney function marker with a normal range of 0.7–1.2 mg/dL). A value of 3.2 mg/dL is nearly three times the upper limit, suggesting significant kidney dysfunction. However, this clinical meaning is not accessible without domain expertise. Hence, we pair SAEs with an LLM-based interpretation pipeline that resolves features to human-readable descriptions using MIMIC-IV (Johnson et al., 2023) reference tables and prompt an LLM to identify shared clinical patterns. This yields natural-language feature descriptions derived from real-world patient data. Together, these components enable the first systematic interpretability analysis of EHR-FMs. Our contributions are as follows:

1. **Cross-system syndrome discovery.** SAE features encode candidate matches to known clinical syndromes — including Cardiorenal, Hepatorenal, Systemic Inflammatory Response Syndrome (SIRS), Urinary tract infection (UTI), Acute Myocardial Infarction (AMI) and Diabetic Ketoacidosis (DKA) as single monosemantic features, unifying conditions that ICD distributes across separate chapters. Named-syndrome matches concentrate in deeper layers for medical conditions (pancreatitis, AMI at Layer 5) while

[1] Department of Radiation Oncology, Mayo Clinic, Rochester, MN, USA. Correspondence to: Shashank Yadav <yadav.shashank@mayo.edu>.

*Proceedings of the $2^{nd}$ ICML Workshop on Foundation Models for Structured Data*, Seoul, South Korea. 2026. Copyright 2026 by the author(s).

others appear earlier (UTI, DKA at Layer 1). From self-supervised learning alone, the EHR-FM organizes clinical concepts along an axis distinct from ICD's anatomy/etiology hierarchy.

2. **Evidence that SAE features mediate computation in the model.** Ablation of a single Layer 1 feature produces >5% downstream changes in a sparse set of Layer 3/5 features for 69.7% of the 400 source–target pairs tested, revealing information-flow circuits through which raw lab signals compose into syndromes layer by layer. Feature ablation provides interventional evidence that SAE-derived features have computational roles in the model.

## 2. Related Work

**EHR Foundation Models**. Transformer-based models pre-trained on longitudinal clinical data have demonstrated strong few-shot transfer to downstream clinical tasks (Wornow et al., 2023; Steinberg et al., 2023). The Medical Event Data Standard (MEDS) (Oufattole et al., 2024) has standardized the representation of clinical events as timestamped code-value pairs, enabling reproducible model training across different EHRs. Despite their predictive success, interpretability methods for these models have remained limited to post-hoc approaches such as attention visualization (Li et al., 2020; Rasmy et al., 2021), gradient-based attribution (Deznabi et al., 2021) and SHAP (Lundberg & Lee, 2017; Deznabi et al., 2021). These methods explain individual predictions but do not reveal what the model has learned as a whole. Their outputs (importance scores or heatmaps) over input codes are difficult to map onto clinically meaningful ontologies, limiting their utility for domain experts (Wysocki et al., 2023).

**SAEs for Mechanistic Interpretability**. Dictionary learning with SAEs was proposed as a principled method for understanding neural network representations (Cunningham et al., 2023; Bricken et al., 2023). A key insight, demonstrated in toy settings by (Elhage et al., 2022), is that neural networks represent more features than they have dimensions (superposition) and SAEs can recover these features by learning an overcomplete basis. TopK SAEs introduced by Gao et al. (2025) enforce exact sparsity ($k$ active features per input) without the need for sparsity penalty tuning. SAEs have also been applied to medical *text* and *vision–language* models: Renzulli et al. (2025) dissect MedCLIP chest-radiograph representations and Modi et al. (2026) train SAEs on MIMIC-IV clinical notes to analyze MedGemma-27B-Text-IT. Complementing dictionary learning, Marinescu et al. (2025) use activation patching, layer lesioning and gradient saliency to map where medical knowledge is stored using techniques that localize knowledge at layer-level resolution but do not decompose representations into individual features. While these methods have advanced interpretability for text-based medical LLMs, none has been applied to EHR-FMs that operate over structured event sequences. We extend SAE-based mechanistic interpretability to EHR foundation models.

## 3. Methods

### 3.1. EHR Foundation Model

We train a 6-layer, 8-head causal transformer (27.5M parameters; $d_{\text{model}}{=}256$, max seq length 512) on MIMIC-IV v3.1 (Johnson et al., 2023) (169M events, 364,627 patients, 80/10/10 patient-level split). Following MEDS-Torch (Oufattole et al., 2024), each event is encoded as a triplet $\mathbf{x}_t = \text{Emb}(\text{code}_t) + \text{CVE}_v(v_t) + \text{CVE}_{\Delta t}(\Delta t_t)$, where CVE refers to Continuous Value Encoders which are linear scalar projections. We use a combined loss function: cross-entropy on next code, MSE on next numeric value, MSE on next log-$\Delta t$. The vocabulary spans 44,524 codes from MIMIC-IV ontologies (28,583 diagnoses, 14,911 procedures, 976 labs, 54 demographic identifiers). Training details are provided in Appendix A.

### 3.2. SAE Training

We train TopK SAEs (Gao et al., 2025) on cached residual-stream activations $\mathbf{h}_\ell \in \mathbb{R}^{256}$ at layers $\ell \in \{1, 3, 5\}$. We run EHR-FM inference over patient trajectories and extract per-event residual states ($\approx$5M vectors per layer). Each SAE (2.1M params) has dictionary size 4,096 with $k{=}32$.

$$\mathbf{z} = \text{TopK}(\mathbf{W}_{\text{enc}}(\mathbf{x} - \mathbf{b}_{\text{dec}}), k), \quad \hat{\mathbf{x}} = \mathbf{W}_{\text{dec}}\mathbf{z} + \mathbf{b}_{\text{dec}}.$$

SAE decoder columns are unit-norm-projected after each step. SAE features are considered alive if they activate above zero on at least one input in the evaluation batch. All three SAEs reach >99% alive features with strongly right-skewed firing-frequency distributions: 75–81% of features fire less often than the uniform-null rate $k/N \approx 8 \times 10^{-3}$, with selectivity increasing in deeper layers (Appendix H)These distributions are inconsistent with indiscriminate firing: activations are gated by recurring input patterns.

### 3.3. LLM-Based Feature Interpretation

We develop an LLM-assisted feature interpretation pipeline: i) for each alive feature, we identify the 30 highest-activating input vectors. ii) trace them back to patient subjects via a subject-to-vector index built during activation caching. iii) extract a 31-event window (15 events before and after) around each activation point. iv) resolve MEDS codes to human-readable descriptions using MIMIC-IV reference tables (e.g., LAB//51221 $\rightarrow$ 'Hematocrit (Blood, Hematology)'). v) prompt an LLM as a judge to identify the common clinical pattern across contexts and return a structured JSON: label, description, category, abstraction level, known

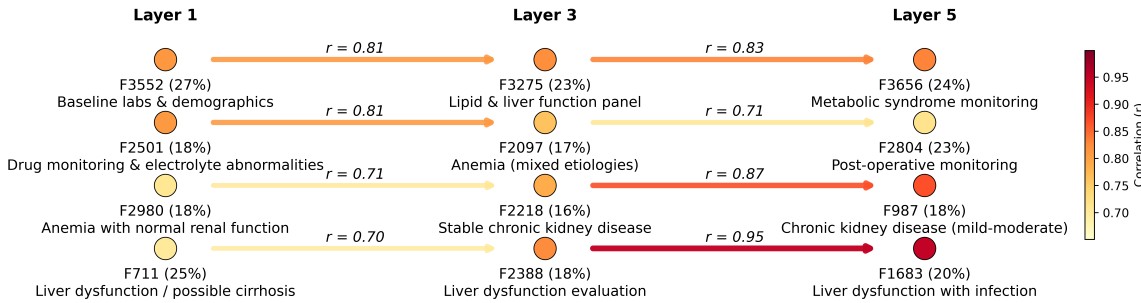

*Figure 1.* **Information flow circuits.** Each row traces a clinical concept as it builds from Layer 1 → Layer 3 → Layer 5. Percentages next to feature IDs denote firing frequency. Interventional validation via activation patching supports these dependencies.

syndrome match and confidence rating. We label 100 features per layer (300 total), yielding 151 high-confidence and 149 medium-confidence interpretations with 17 instances of known (ICD-matched) syndromes. We provide the prompt in Appendix C.

### 3.4. Quantitative Validation

We validate SAE-derived features along three axes. i) *Interventional*: We use activation patching to nullify a decoder direction and measure downstream effects (Appendix E). ii) *External validity*: We use per-subject mean activations to differentiate patients whose ICD codes match the LLM label, yielding a held-out AUROC against ICD phenotypes (Appendix F). iii) *Internal consistency*: a sentence-transformer UMAP of LLM labels checks that features with the same clinical category cluster together (Appendix F).

## 4. Results

### 4.1. Prevalence-Banded Feature Organization

We find that SAE-derived features are organized by activation frequency (prevalence) rather than along ICD's anatomy/etiology hierarchy (Table 1). Frequencies span four to five orders of magnitude and split into four illustrative bands. *Pervasive* features (>30%) capture broad patterns that recur across many admissions, such as altered mental status workups and chronic cardiorenal exacerbations. *Common* features (10–30%) correspond to recurring contexts like pregnancy monitoring or baseline admission labs. *Selective* features (3–10%) and *rare* features (<3%) encode more specific clinical situations such as chronic liver disease with coagulopathy or microcytic anemia evaluation. The same band structure recurs at every layer, indicating that prevalence is a stable organizing axis rather than a layer-specific artifact. Across 300 interpreted features, the LLM proposes 79 candidate matches to named syndromes spanning approximately 17 distinct syndrome categories (Table 2); these matches are concentrated in the *pervasive* and *common* bands, consistent with high-prevalence syndromes being easier to recover from self-supervision alone.

Beyond prevalence, we tested whether layer depth stratifies features by complexity and found that it does not. We categorize the LLM-derived 'abstraction level' field into three tiers: *lab-level* (raw lab patterns), *disease-level* (single-organ or single-disease entities) and *syndromic* (multi-finding clinical scenarios) and tabulate the per-layer proportions at $n$=500 features per layer (Figure 3). The distribution is statistically indistinguishable across Layer 1 (L1), Layer 3 (L3) and Layer 5 (L5) ($\chi^2$ test of independence, $p$=0.56). All three layers carry features at every tier in similar proportions, so model depth does not implement a clean lab-to-syndrome pipeline at the population level. This suggests that cross-layer dependence is carried by specific feature-to-feature connections rather than layer-wide trends.

### 4.2. Cross-Layer Information-Flow Circuits

Cross-layer structure surfaces at the feature-pair level. We compute per-subject mean activations for each feature at L1, L3, L5, then Pearson correlations between all adjacent-layer pairs. Random pairs are essentially uncorrelated across layers (median $r$≈0, 99th-percentile $r$=0.62 for L1→L3 across 50,000 random pairs; Appendix G), so we filter to high-$r$ ($r$>0.7) pairs and trace chains through the three layers. Figure 1 shows representative chains: 'Baseline labs' (L1, $r$=0.81) → 'Lipid+LFTs' (L3, $r$=0.83) → 'Metabolic syndrome' (L5). We use activation patching to distinguish computational dependence from patient-level confounding. We set a L1 feature's decoder direction in the residual stream to zero and measure the downstream change. For instance 'Macrocytic anemia with dysplasia' (L1:F2414) zeroed → $-70\%$ at 'CKD and depression' (L3:F3907). Overall, 69.7% of 400 source–target pairs show >5% downstream effects (Appendix E). We observe that the effects of activation patching are sparse: zeroing a single L1 feature changes a median of 4/100 L3 targets and 0.5/100 L5 targets by $> 5\%$, with 86% of targets remaining within ±1%. The largest effects concentrate on targets whose pre-intervention

correlation with the source exceeds the 99th percentile of random pairs (Appendix G), so patching effects align with the high-correlation pathways used to identify the circuits. Random L1→L5 pairs at matched correlation value produce only minor downstream changes when patched (median $|\Delta| < 0.2\%$, vs. 2.0% for labeled circuits at the same r level). This suggests that generic cross-layer coupling at high r is not sufficient to produce the labeled effects.

### 4.3. Categorical Organization of Feature Labels

Figure 2 shows a UMAP (McInnes et al., 2018) projection of feature label embeddings, colored by clinical category. Demographic features form an isolated cluster in the upper region of the projection and renal, hematological and gastrointestinal features each occupy separate regions. Categories with broader clinical scope behave consistently with this: multi-system and metabolic features overlap centrally and infectious features are distributed across the projection, consistent with their cross-organ manifestations. This indicates that the LLM-assigned labels are internally consistent: features the LLM groups into the same clinical category have semantically similar descriptions.

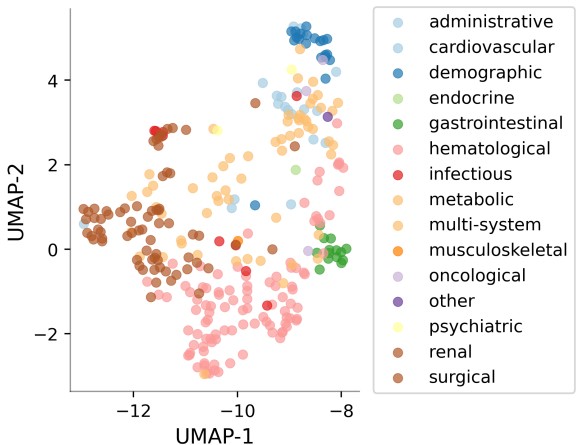

*Figure 2.* UMAP of all 300 LLM-assigned feature labels (sentence-transformer embeddings, cosine metric). Colored by clinical category. Features with the same category form coherent clusters across all three layers, indicating internally-consistent labeling.

## 5. Discussion

The sparse cross-layer correlation structure (Figure 1) and uniform abstraction-tier distribution across layers (Figure 3) indicate that the model uses its depth through a small set of feature-to-feature pathways rather than as a uniform complexity hierarchy. Activation patching shifts downstream features for tested L1→L3/5 pairs, providing interventional rather than purely correlational evidence that specific features compose into syndromic ones.

The learned representations differ from ICD along two axes.

Cross-system syndromes such as cardiorenal disease and DIC appear as single features rather than codes split across chapters. Feature organization is based on activation prevalence rather than anatomy or etiology. Our results suggest the model captures both canonical clinical knowledge and patterns specific to the source of training data. The model encodes named clinical syndromes as single monosemantic features, consistent with rediscovery of canonical knowledge, but it also encodes patterns that have no ICD analog. Data from a second site would be needed to separate canonical knowledge from site-specific patterns.

Strong mechanistic interpretability would require features to be meaningful across retraining and different training data. SAE features meet several of these criteria: the dictionary recovers named clinical syndromes, groups features by category and supports interventional analysis through activation patching. However, portability of SAE features across checkpoints remains open.

**Limitations:** The foundation model is trained on a single EHR dataset, so we cannot distinguish features that capture universal clinical patterns from those that reflect institution-specific ordering conventions. Multi-site replication on a second EHR would identify which features transfer and which are institution-specific. The interpretation analysis covers a sample of features per layer rather than the full dictionary and scaling interpretation to all alive features would test whether the prevalence bands and syndrome counts hold at full coverage. Feature labels depend on both the interpretation prompt and the judge LLM, so we treat them as candidate hypotheses rather than definitive descriptions. Also, a broader inter-rater comparison against stronger frontier LLMs is precluded by the PhysioNet data use agreement, which prohibits transmit- ting MIMIC-IV records to third-party LLMs. Finally, activation patching here is restricted to single-feature interventions and ablating syndromic features in matched patient cohorts to measure downstream prediction changes is left to future work.

## 6. Conclusion

We show that sparse autoencoders with LLM-based labeling can reveal candidate clinical concepts in an EHR foundation model. The EHR-FM organizes clinical concepts in a structure distinct from ICD through self-supervised learning, with features organized by prevalence. ICD-matched syndromes appear as single monosemantic features and cross-layer chains emerge after correlation-based filtering. Activation patching provides interventional evidence consistent with features having computational roles in the foundation model. We show SAE-based mechanistic interpretability is a useful approach for understanding EHR foundation models, with implications for ontology development from large clinical datasets.

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

# A. EHR-FM Training

## A.1. MEDS representation

The raw data source is MIMIC-IV v3.1 (Johnson et al., 2023). We convert it to the Medical Event Data Standard (MEDS) format (Oufattole et al., 2024) using the MEDS-ETL pipeline. Each row is one clinical event with columns: i) `subject_id` (int64): Patient identifier. ii) `time` (datetime): Event timestamp. Static events (sex, date of birth) carry a null time. iii) `code` (str): Hierarchical code string, for example `LAB//51221` (hematocrit), `DIAGNOSIS//ICD//10//I50.9` (heart failure), `PROCEDURE//ICD//9//88.54`, or `DEMOGRAPHIC//GENDER//F`. iv) `numeric_value` (float32, nullable): Value for lab/vital events; NULL for events without a value. A dictionary enumerates every code that appears in the cohort and assigns it an integer. Index 0 is reserved for padding. The dataset is split by `subject_id` without leakage: train (291,701 patients, 135M events), validation (36,462 patients, 17M events), held-out (36,464 patients, 17M events).

## A.2. Tokenization and input encoding

The dataset loader groups events by subject ID, sorts them by time, chunks into non-overlapping windows of 512 events (minimum 4 events per window) and returns four parallel per-event arrays: integer code indices, z-score-normalized numeric values, $\log(1 + \Delta t_{\text{days}})$ time deltas and a boolean padding mask. The model converts these to per-position $d_{\text{model}}$-dimensional input vectors as the sum of three independently-learned embeddings:

$$\mathbf{x}_t = E_c(\text{code}_t) + \text{CVE}_v(v_t) + \text{CVE}_{\Delta t}(\Delta t_t), \quad (1)$$

where $E_c \in \mathbb{R}^{(|V|+1) \times 256}$ is the code embedding and each continuous-value encoder is a single linear layer mapping $\mathbb{R}^1 \to \mathbb{R}^{256}$ applied to the scalar. The input is then passed through an input layer normalization and dropout before entering the transformer stack.

## A.3. Transformer architecture

The encoder is a stack of 6 RoPE (Su et al., 2024) transformer blocks, each implementing: i) $d_{\text{model}} = 256$, 8 attention heads ($d_{\text{head}} = 32$), $d_{\text{ff}} = 1024$, dropout 0.1; ii) pre-LayerNorm; separate $W_Q, W_K, W_V, W_{\text{out}}$ linear projections; rotary positional embeddings applied to $Q$ and $K$ after per-head reshape; iii) a causal $-\infty$ triangular mask plus an additive $-\infty$ mask for padded key positions, both combined into a single attention mask; iv) a post-attention MLP (Linear → GELU → Dropout → Linear → Dropout), residual connections at both sublayers. After the stack, a final layer normalization is applied. Three linear heads then operate on every position: a code head $\in \mathbb{R}^{256 \to |V|+1}$, a value head $\in \mathbb{R}^{256 \to 1}$ and a time head $\in \mathbb{R}^{256 \to 1}$.

## A.4. Training objective and procedure

The loss combines three terms:

$$\mathcal{L}(\theta) = \mathcal{L}_{\text{code}} + \alpha_v \, \mathcal{L}_{\text{value}} + \alpha_t \, \mathcal{L}_{\text{time}}, \quad \alpha_v = \alpha_t = 0.1,$$

with

$$\mathcal{L}_{\text{code}} = -\frac{1}{|\mathcal{T}_{\text{code}}|} \sum_{t \in \mathcal{T}_{\text{code}}} \log p_\theta(\text{code}_{t+1} \mid \cdot),$$

$$\mathcal{L}_{\text{value}} = \frac{1}{|\mathcal{T}_v|} \sum_{t \in \mathcal{T}_v} (\hat{v}_{t+1} - v_{t+1}^\star)^2,$$

$$\mathcal{L}_{\text{time}} = \frac{1}{|\mathcal{T}_\tau|} \sum_{t \in \mathcal{T}_\tau} (\hat{\tau}_{t+1} - \tau_{t+1}^\star)^2,$$

where $\mathcal{T}_{\text{code}}$ excludes padding, $\mathcal{T}_v$ further restricts to positions whose target normalized value is non-zero and $\mathcal{T}_\tau$ excludes padding. Value targets use training-set z-score normalization; time targets use $\log(1 + \Delta t_{\text{days}})$. We use the Adam optimizer (Kingma & Ba, 2014) with learning rate $3 \times 10^{-4}$, linear warmup + cosine-annealing schedule, per-step batch size 64 with gradient accumulation. We use BF16 mixed precision training with gradient clipping to global norm 1.0 and early stopping with patience 10, up to 50 epochs.

## A.5. Activation caching

We extract residual-stream activations at layers $\ell \in \{1, 3, 5\}$ from the frozen, trained EHR-FM via a forward hook registered on the corresponding layer. The hook stores the layer's output tensor *after* the complete RoPE transformer block with shape `[batch, seq_len, 256]`. For each batch, we mask out padding positions and keep only per-event vectors at valid positions. We also track the originating patient's subject ID and the number of vectors contributed by that patient. We save approximately 5M vectors per layer ($\approx$5.1 GB per layer in float32).

# B. SAE training

The trained EHR-FM is run over each patient's trajectory; padding is discarded; valid per-event vectors are flattened and concatenated into a tensor of shape $\mathbb{R}^{N \times 256}$ ($N \approx$ 5M per layer, from $\approx$250K training-split patients). Each optimizer step draws a uniformly random batch of 4,096 vectors from this pool, mixing events from thousands of patients per batch.

Each TopK SAE has $W_{\text{enc}} \in \mathbb{R}^{4096 \times 256}$, $W_{\text{dec}} \in \mathbb{R}^{256 \times 4096}$ with unit-norm-projected columns and pre-encoder bias subtraction. TopK activation retains the $k = 32$ largest entries with ReLU, zeroing the rest. SAE training uses the Adam optimizer with learning rate $3 \times 10^{-4}$, batch 4,096, 5 epochs ($\approx$6,100 steps), cosine annealing. The 5M-vector pool is used end-to-end without a held-out split, so alive/dead/$\ell_0$

stats are training-set numbers. Downstream claims (feature interpretation, patching, label-validation AUROC) are computed across the full patient cohort and so generalize beyond this pool.

We also observe that the reconstruction loss rises with depth, consistent with later-layer activations being harder to reconstruct at fixed $k$ and dictionary size. The percentage of alive features rises from 99.1% to 100%, indicating that few features are permanently dead. Feature entropy peaks at Layer 3, consistent with the observation that middle layers carry the most distributed representations in language-model interpretability (Templeton, 2024) and we treat this as a descriptive statistic rather than evidence of a specific mechanistic claim.

## C. LLM Interpretation Pipeline Details

For each of the 300 analyzed features (100 per layer), the pipeline:

1. **Selects top-activating vectors.** From the 5M cached SAE activations per layer, identifies the 30 vectors with the highest activation for that feature.

2. **Traces back to patients.** Each activation vector is mapped back to a specific patient and position within their clinical event sequence via an explicit subject-to-vector index.

3. **Extracts context windows.** For each of the 30 top-activating patients, extracts 15 events before and 15 events after the activation point (31 events per patient).

4. **Resolves clinical codes.** Raw MEDS codes (e.g., a hematocrit lab code) are translated to human-readable descriptions (e.g., "Hematocrit (Blood, Hematology)") using MIMIC-IV reference tables (d_labitems, d_icd_diagnoses, d_icd_procedures).

5. **Prompts the LLM.** A single prompt containing ~30 patient contexts (~600 clinical events total) is sent to the LLM, which identifies the common clinical pattern and returns a structured response: label, description, category, abstraction level, known syndrome match and confidence rating. The prompt is provided verbatim below.

```
You are a clinical informatics
expert analyzing features learned
by a sparse autoencoder (SAE)
applied to an EHR (Electronic Health
Record) transformer model trained
on MIMIC-IV (Beth Israel Deaconess
Medical Center ICU data, 364,627
patients).

I will show you the clinical event
sequences that most strongly activate
a specific SAE feature.  Each
example shows the events surrounding
the activation point (marked with
>>>).

Your task:

1.  Identify the common clinical
pattern across ALL examples --- look
for what is SHARED, not what varies

2.  Determine the ABSTRACTION
LEVEL: is this a raw lab pattern, an
organ-system condition, or a complex
multi-system clinical scenario?

3.  If this pattern matches a known
clinical syndrome or phenotype, name
it (e.g., cardiorenal syndrome,
SIRS, DIC, hepatorenal syndrome,
metabolic syndrome)

4.  Rate your confidence (high/medium/low)
based on pattern consistency

Respond in this exact JSON format:
{ "label":  "Short label (3-8 words,
e.g., 'Acute kidney injury with
dialysis')", "description":  "2-3
sentence clinical description.  If
this matches a known syndrome, state
which one.", "clinical_category":
"One of:  cardiovascular, respiratory,
renal, metabolic, infectious,
neurological, hematological, gastrointestinal,
psychiatric, oncological, surgical,
demographic, multi-system, temporal,
administrative, other", "abstraction_level":
"One of:  lab_pattern, organ_system,
clinical_condition, clinical_scenario",
"key_indicators":  ["list", "of",
"key", "clinical", "events"],
"known_syndrome":  "Name of matching
clinical syndrome if applicable,
otherwise null", "confidence":
"high/medium/low", "reasoning":
"Brief explanation of why you chose
this interpretation" }

Here are the {n_examples} highest-activating
clinical contexts for Feature
{feature_idx}:

{contexts}
```

The entire interpretation is done on a locally hosted LLM[1] (Qwen 2.5 7B). The LLM sees 30 patients per feature, those that most strongly activate it. The LLM is shown 15 events before and 15 events after each activation point. The feature itself depends only on causal context, since the

---

[1]MIMIC-IV is credentialed data under the PhysioNet Data Use Agreement, which prohibits transmitting records to third-party services such as commercial LLM APIs; locally hosted, open-weight models that run on local hardware are permitted. See PhysioNet, "Responsible use of MIMIC data with online services like GPT," https://physionet.org/news/post/llm-responsible-use/.

*Table 1.* Illustrative prevalence bands among LLM-interpreted features. Cut-points are log-spaced ($\sim 3\times$ between bands) since activation frequency spans 4–5 orders of magnitude.

| Band | Frequency | Example features (LLM-interpreted) |
|------|-----------|-----------------------------------|
| Pervasive | >30% | "Altered mental status admission" (48%), "Chronic cardiorenal exacerbation" (42%), "Macrocytic anemia with dysplasia" (36%) |
| Common | 10–30% | "Pregnancy monitoring" (28%), "Baseline labs and demographics" (27%), "Anemia + AKI" (27%) |
| Selective | 3–10% | "Chronic liver disease with coagulopathy" (10%), "Abnormal RBC morphology" (10%) |
| Rare | <3% | "Possible UTI" (3%), "Microcytic anemia evaluation" (3%) |

*Table 2.* Candidate SAE-feature matches to named clinical syndromes, assigned by LLM labeling from top-activating patient contexts (see Appendix C for criteria and prompt). "Discovered without syndrome-level supervision" refers to the fact that the model is trained only with self-supervised next-event prediction; the candidate labels themselves come from the LLM and are prompt-sensitive.

| Syndrome | Feature | Layer | Freq |
|----------|---------|-------|------|
| Cardiorenal Syndrome | F398 | 5 | 42% |
| Hepatorenal Syndrome | F3404 | 3 | 10% |
| Portal Hypertension | F1549 | 5 | 42% |
| Metabolic Syndrome | F3767 | 3 | 9% |
| SIRS | F2948 | 5 | 15% |
| DIC | F3478/F462 | 5/1 | 8% |
| Acute MI | F3953 | 5 | 12% |
| DKA | F3375 | 1 | 10% |
| Gestational DM | F3605 | 3 | 7% |
| Pancreatitis | F2529 | 5 | 10% |

residual-stream vector at event t is produced from events before that timepoint; post-hoc events are shown to help the LLM identify the activating pattern, not to alter what the feature encodes. Throughout the paper, "known clinical syndrome" refers to a named multi-finding clinical entity that is either (a) a recognized syndrome with an established definition (e.g., SIRS, DIC, cardiorenal syndrome, hepatorenal syndrome, metabolic syndrome), or (b) a classical single-disease entity with a standard diagnostic pattern (e.g., DKA, pancreatitis, AMI, UTI). A feature is counted as a "candidate match" to one of these when the LLM names that entity as the known syndrome and the confidence is medium or high.

# D. Extended Results

## D.1. Syndrome Discovery and Layer-Wise Emergence

We list the layer at which each candidate syndrome feature was identified within the 100-feature-per-layer interpretation sample (Table 3). Within the 100-feature-per-layer sample, single-disease patterns such as UTI, DKA, and cirrhosis appear at Layer 1, while multi-finding syndromes such as cardiorenal, hepatorenal, AMI, and pancreatitis appear predominantly at Layer 3 or Layer 5. Several syndromes appear at multiple layers (CKD, metabolic syndrome, cardiorenal), suggesting that related features at different depths may track the same clinical concept at different levels of abstraction.

## D.2. Abstraction-Level Distribution Across Layers

We collapse the LLM's abstraction-level field into three tiers to check whether feature complexity varies systematically across layer indices: *lab-level* (lab pattern), *disease-level* (organ system + clinical condition), *syndromic* (clinical scenario) and tabulate the per-layer share (Figure 3). Table 4

*Table 3.* Layer at which each candidate syndrome feature was identified within the 100-feature-per-layer interpretation sample. ✓ indicates a candidate match found at that layer; "—" indicates no candidate match within the sampled features at that layer (not a guarantee that no such feature exists).

| Syndrome | L1 | L3 | L5 |
|----------|----|----|----|
| UTI | ✓ | ✓ | — |
| DKA | ✓ | — | — |
| CKD | ✓ | ✓ | ✓ |
| Diabetes | ✓ | ✓ | — |
| Cirrhosis | ✓ | — | — |
| Metabolic Syndrome | ✓ | ✓ | ✓ |
| Cardiorenal | ✓ | ✓ | ✓ |
| Hepatorenal | — | ✓ | ✓ |
| Metabolic Acidosis | — | ✓ | ✓ |
| MDS | — | ✓ | — |
| Gestational DM | — | ✓ | — |
| CHF | — | ✓ | — |
| SIRS | ✓ | — | ✓ |
| DIC | ✓ | — | ✓ |
| AMI | — | — | ✓ |
| Pancreatitis | — | — | ✓ |

gives the underlying counts at $n{=}500$ per layer.

Two points to note: (i) Lab-pattern features are present at every layer, including L5 ($\sim$42%), so raw-signal-style features are not confined to early layers. (ii) Clinical-scenario features are present at every layer, including L1 ($\sim$20%); many of the L1 scenarios are generic 'multi-system lab abnormalities' rather than candidate matches to named syndromes. The narrower layer-emergence pattern that does survive is at the level of *named* clinical syndromes (Table 3): pancreatitis, AMI and cardiorenal-as-named appear only at L5; UTI and DKA at L1.

*Table 4.* Counts of features at each tier per layer at the $n{=}500$ sample depth. A chi-square test of independence between tier and layer is non-significant ($\chi^2{=}3.00$, dof=4, $p{=}0.56$); a Cochran-Armitage trend test of multi-system share across L1→L3→L5 is also non-significant ($Z{=}-0.83$, $p{=}0.41$).

| Tier ($n{=}500$ per layer) | L1 | L3 | L5 | Total |
|---|---|---|---|---|
| lab-level (lab pattern) | 213 | 219 | 208 | 640 |
| disease-level (organ/condition) | 189 | 202 | 204 | 595 |
| syndromic (scenario) | 98 | 79 | 88 | 265 |

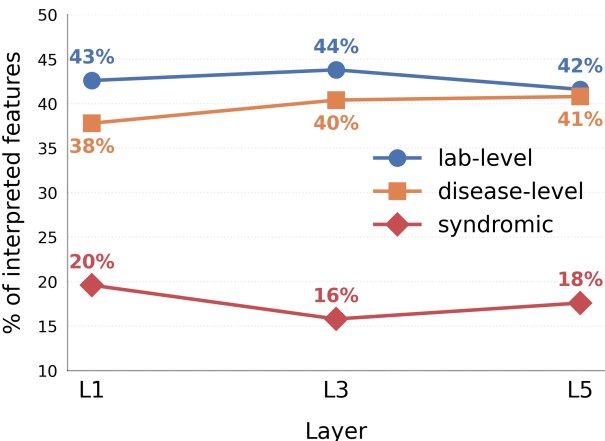

*Figure 3.* LLM-assigned abstraction tier across layers ($n{=}500$ features per layer). All three tiers are present at every layer in similar proportions

## E. Activation Patching

We perform activation patching to probe whether the cross-layer correlations in Figure 1 involve direct computational dependence between features rather than being sustained by shared patient-level covariates: for each top Layer 1 feature we zero out its decoder-direction contribution to the residual stream and measure the resulting change in Layer 3 and Layer 5 feature activations. Figures 4a–4b show the patching effect matrices. These matrices have sparse structure: zeroing a single Layer 1 feature produces large changes in a small number of downstream features and leaves most others largely unchanged. This pattern is consistent with targeted information flow mediated by the SAE features. Of the 400 source–target pairs tested, 69.7% show measurable downstream effects (>5% activation change).

## F. LLM Label Validation Details

We validate the LLM-assigned labels with two complementary checks: an *external-validity* test (do labels predict the ICD codes they ought to track at the population level?) and an *internal-consistency* test (do features with similar LLM labels cluster together in label space?).

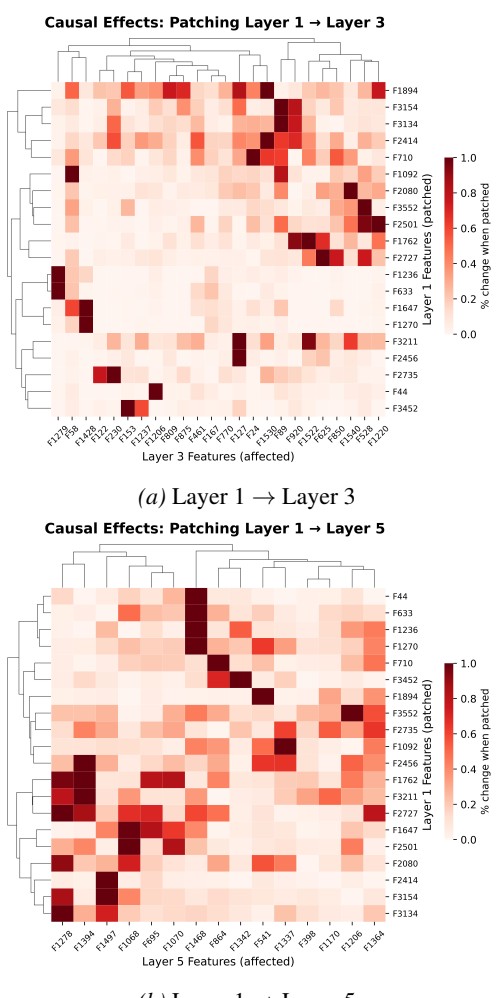

*(a)* Layer 1 → Layer 3

*(b)* Layer 1 → Layer 5

*Figure 4.* Activation patching heatmaps (clustered). Each cell shows the % change in a downstream feature when a Layer 1 feature is zeroed. Block structure reveals groups of features connected by activation patching. Feature IDs are mapped to interpretations in Table 5- 7.

### F.1. External validity: held-out ICD AUROC

We test whether LLM-assigned labels have any systematic association with ICD-level phenotypes. For each interpreted feature, we:

i) Extract the LLM's label and description (e.g., 'Chronic cardiorenal disease exacerbation'). ii) Map the label to ICD code prefixes via a lookup of syndrome categories (e.g., 'cardiorenal' → {I50, N17, N18, I13}). iii) For *every* patient in the dataset, compute: (a) mean activation of this feature across all their clinical events and (b) a binary label indicating whether any of their ICD codes match the syndrome's codes. iv) Compute AUROC for whether the feature's activation discriminates patients with matching ICD codes from those without. Only 90/300 features have ICD-matchable labels; the rest encode patterns without direct ICD equiv-

alents (workflow, post-op monitoring). A mean AUROC of 0.541 is modest but expected, since SAE features capture multi-finding patterns that ICD codes only coarsely approximate. We observe that 79% of these 90 features (71/90) score strictly above 0.5, vs. 50% under the random-ordering null ($p < 2 \times 10^{-5}$, one-sided binomial). This pattern reflects systematic but modest alignment between LLM labels and ICD-level phenotypes. All external validation here is performed within MIMIC-IV, so the held-out ICD phenotypes share the same data-generating distribution as the SAE training data. Other ICU EHR datasets (e.g., eICU-CRD) have been shown to be out-of-distribution relative to MIMIC-IV (Yadav, 2025), so future work should test whether this feature-label validation holds under cross-dataset evaluation.

### F.2. Internal consistency: label-space UMAP

We embed the (label + description) text of all 300 features with a sentence transformer (all-MiniLM-L6-v2; Reimers & Gurevych, 2019) and project to 2D via UMAP (McInnes et al., 2018) (15 neighbors, minimum distance 0.1, cosine metric) to examine whether LLM-assigned labels are internally consistent rather than noise. Features with the same LLM-assigned category form largely separable clusters (hematological, renal, gastrointestinal, demographic/administrative, multi-system) as illustrated in Figure 2.

## G. Cross-Layer and Co-activation Structure

We compute Pearson correlations between per-subject mean feature activations across layers (Figure 5). Block structure shows groups of early features that jointly predict specific late-layer phenotypes (e.g., an L1 cluster of creatinine/BUN/electrolyte features maps onto a single L5 cardiorenal feature). Within Layer 3, hierarchical clustering on the co-activation matrix (top 50 features) yields clusters corresponding to hematological, renal, hepatic and metabolic themes. As a sparsity check we sample 50,000 random feature pairs across each layer pair (drawing source and destination indices uniformly from features with non-trivial variance) and plot the distribution of Pearson correlations (Figure 6). The bulk of the distribution is centered near zero (median $r = -0.030$ for L1→L3, $-0.023$ for L3→L5, $-0.027$ for L1→L5); the 99th percentile is 0.62, 0.54 and 0.56 respectively. The ten labeled circuit pairs (red lines) all sit in the right tail with $r$ between 0.58 and 0.95, i.e. at or above the 99th percentile of random pairs. This is consistent with the cross-layer connections being sparse: typical feature pairs across layers are uncorrelated, while the labeled chains lie at the high-$r$ extreme of the distribution.

## H. Active-Feature Firing Distribution

For each of the 4,096 features per layer we plot the activation frequency, i.e., the fraction of cached input event-vectors at which the feature is non-zero (Figure 7). TopK SAEs constrain exactly $k=32$ of 4,096 features to fire per input, so under uniformly random feature firing every feature would have frequency exactly $k/N=32/4096\approx0.0078$. Departures from this null are evidence that features have specialized roles. The empirical distributions are heavy-tailed and bimodal across all three layers: a left peak of rare features (frequency $10^{-5}$–$10^{-4}$, well below the null) and a right peak of frequent "context" features (frequency $10^{-2}$–$10^{-1}$, well above the null), with relatively few features sitting at the null itself. Alive feature counts are 4,059/4,092/4,096 at L1/L3/L5 ($\geq$99.1%); the share firing on >1% of inputs is 20.1/18.9/17.7%; the share firing on >10% is 1.2/1.1/1.6% (no random allocation could place any feature above 10% by chance). Median firing frequency among alive features drops $\sim$200$\times$ from L1 ($1.4\times10^{-3}$) to L5 ($7.0\times10^{-6}$), so a typical L5 feature fires on far fewer inputs than the typical L1 feature.

## I. Data and Code Availability

We use MIMIC-IV v3.1 (Johnson et al., 2023), available through PhysioNet under a credentialed data use agreement. Code is available at `https://github.com/xinformatics/sae-clinical`.

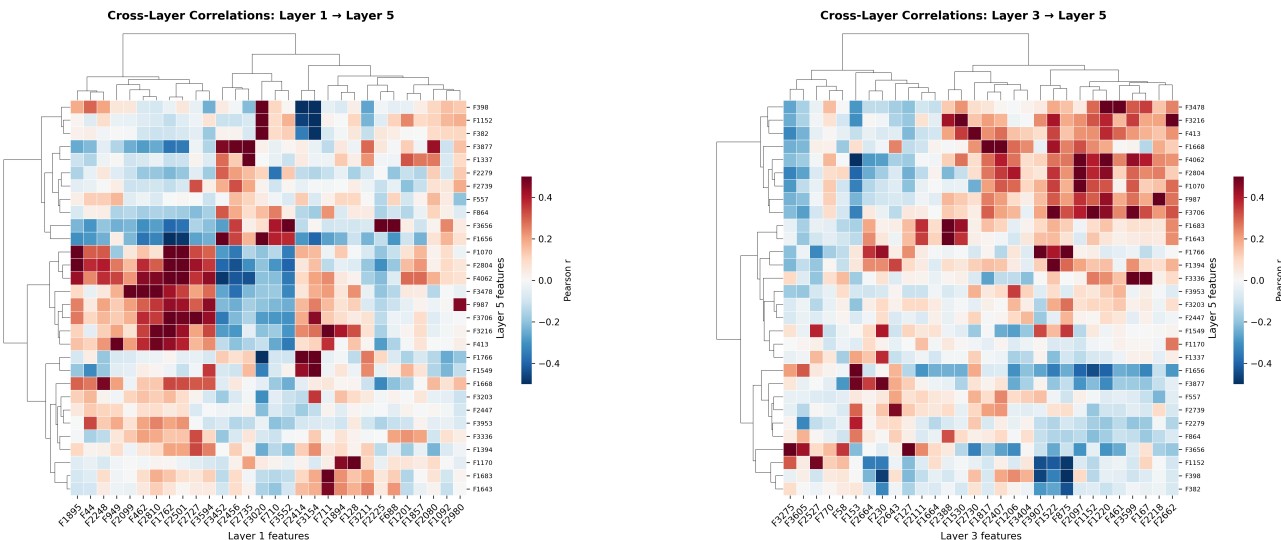

*Figure 5.* Cross-layer correlations: L1→L5 (left, max $r$=0.74) and L3→L5 (right, max $r$=0.95).

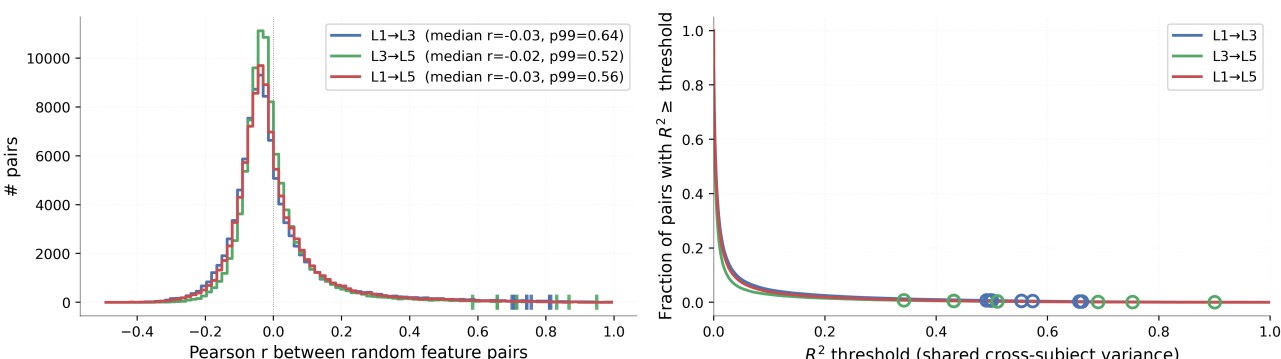

*Figure 6.* Random cross-layer feature-pair correlations (50,000 pairs/layer-pair). Left: Signed r histogram (log y); colored ticks: labeled circuits from Figure 1. Median r ≈ 0 across all layer-pairs. Right: R² survival curve, log y; rings mark labeled circuits at (R², random-pair tail mass). Every labeled chain sits at R² ≥ 0.34, a level reached by less than 1% of random pairs.

*Table 5.* Feature ID lookup (Layer 1). All feature IDs appearing in Figures 4a–4b with LLM-assigned labels and syndrome matches. Sorted by activation frequency. Syndrome matches: [a]Cirrhosis, [b]UTI, [c]DIC, [d]DKA.

| ID | LLM Interpretation | Category | Freq | ID | LLM Interpretation | Category | Freq |
|---|---|---|---|---|---|---|---|
| F3211 | Anemia with dysplastic blood features | Hematological | 39% | F1895 | Post-operative anemia, electrolyte changes | Surgical | 18% |
| F2414 | Macrocytic anemia with dysplasia | Hematological | 36% | F2501 | Therapeutic drug monitoring | Multi-system | 18% |
| F2456 | Warfarin/Coumadin therapy monitoring | Hematological | 32% | F2980 | Anemia with normal renal function | Renal | 18% |
| F1092 | Coagulation abnormalities, liver dysfunction | GI | 31% | F2225 | Screening for chronic infections | Infectious | 18% |
| F44 | Macrocytic anemia evaluation | Hematological | 30% | F688 | Anemia with variable cytopenias | Hematological | 17% |
| F1894 | Elevated uric acid with variable findings | Renal | 30% | F2727 | Anemia with chronic kidney disease | Renal | 17% |
| F710 | Pregnancy confirmation and monitoring | Demographic | 28% | F3020 | Neutrophilia with possible infection | Infectious | 17% |
| F1762 | Anemia with Vancomycin/Gentamicin use | Hematological | 27% | F462[c] | Complex coagulopathy, multi-organ involvement | Multi-system | 16% |
| F3552 | Baseline labs and demographics | Demographic | 27% | F949 | Mild liver dysfunction with normal labs | GI | 15% |
| F3154 | Anemia with renal involvement | Renal | 27% | F128 | Anemia with bone marrow involvement | Hematological | 14% |
| F711[a] | Liver dysfunction, possible cirrhosis | GI | 25% | F3134 | Variable anemia and WBC changes | Hematological | 14% |
| F2735 | Anemia with variable WBC counts | Hematological | 22% | F1647 | Renal insufficiency with possible infection | Renal | 13% |
| F2080 | Routine lab testing with thyroid assessment | Administrative | 21% | F3375[d] | Metabolic emergency with acidosis | Metabolic | 10% |
| F3452[b] | Urinary tract infection evaluation | Renal | 21% | F1270 | Complex chronic illness with anemia | Multi-system | 5% |

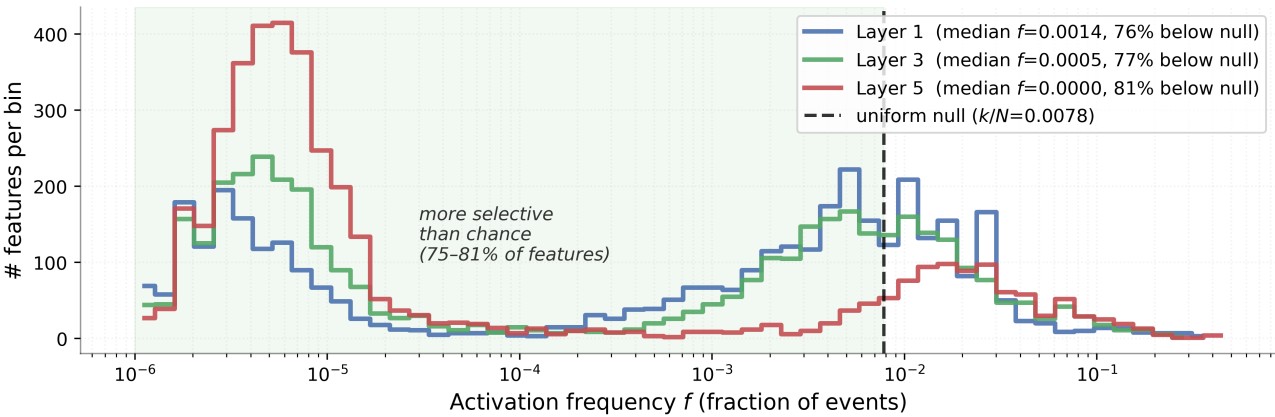

*Figure 7.* Per-feature activation frequency by layer. The black dash-dotted line at $f=32/4096\approx0.0078$ marks the uniform-null expectation under random feature firing; under random allocation every feature would sit at this line. Empirical distributions are bimodal and span four orders of magnitude, a left peak of rare specialized features and a right peak of high-prevalence "context" features.

*Table 6.* Feature ID lookup (Layer 3). Continued from Table 5. Syndrome matches: [a]Diabetes, [b]Hepatorenal, [c]Metabolic Acidosis, [d]MDS, [e]Gestational Diabetes Mellitus.

| ID | LLM Interpretation | Category | Freq | ID | LLM Interpretation | Category | Freq |
|---|---|---|---|---|---|---|---|
| F1817 | Anemia with variable thrombocytopenia | Hematological | 26% | F1530[b] | Cirrhosis with complications | GI | 16% |
| F2664[a] | Diabetes mellitus screening | Metabolic | 25% | F2643 | Anemia evaluation with macrocytosis | Hematological | 16% |
| F2527 | Macrocytic anemia, marrow involvement | Hematological | 24% | F1152 | Routine lab testing with comorbidities | Multi-system | 16% |
| F3275 | Lipid panel with liver function tests | Metabolic | 23% | F875 | Admission with multiple comorbidities | Multi-system | 15% |
| F153 | CBC and chemistry panel order | Administrative | 22% | F1220[c] | Mixed acid-base disorder | Renal | 15% |
| F1206 | Anemia evaluation, normal chemistries | Hematological | 22% | F2407 | Anemia with normal renal function | Hematological | 15% |
| F3907 | Chronic kidney disease and depression | Renal | 22% | F1664 | Anemia with possible infection | Hematological | 15% |
| F2730 | Normal INR/PT with prolonged PTT | Hematological | 19% | F230 | Female gender with routine labs | Demographic | 15% |
| F2111 | Anemia evaluation, variable findings | Hematological | 18% | F461 | Renal dysfunction and anemia evaluation | Renal | 15% |
| F2388 | Mild liver dysfunction evaluation | GI | 18% | F127 | Normal renal function with routine labs | Renal | 14% |
| F3599 | Complex anemia evaluation | Hematological | 18% | F167 | Anemia with renal/urinary abnormalities | Renal | 14% |
| F770 | Anemia with renal dysfunction | Renal | 18% | F3404[b] | Hepatorenal syndrome with anemia/AKI | Multi-system | 13% |
| F58 | Thyroid dysfunction and related testing | Endocrine | 17% | F2662[d] | Anemia with signs of myelodysplasia | Hematological | 12% |
| F1522 | Urine abnormalities, kidney dysfunction | Renal | 17% | F3605[e] | Anemia in pregnancy with glucosuria | Renal | 12% |
| F2097 | Anemia with possible blood loss | Hematological | 17% | F850 | Chronic liver disease with coagulopathy | GI | 10% |
| F2218 | Stable chronic kidney disease | Renal | 16% | | | | |

*Table 7.* Feature ID lookup (Layer 5). Continued from Table 6. Syndrome matches: [a]Cardiorenal, [b]Hepatorenal, [c]CKD, [d]AMI, [e]DIC.

| ID | LLM Interpretation | Category | Freq | ID | LLM Interpretation | Category | Freq |
|---|---|---|---|---|---|---|---|
| F1766 | Admission with altered mental status | Multi-system | 48% | F3877 | Joint inflammation, possible infection | Musculoskeletal | 19% |
| F398[a] | Chronic cardiorenal disease exacerbation | Multi-system | 42% | F4062 | Anemia with variable organ involvement | Hematological | 19% |
| F1549[b] | Complex liver disease, thrombocytopenia | Multi-system | 42% | F557 | Routine lab testing, gender marker | Demographic | 18% |
| F1152 | Macrocytic anemia workup | Hematological | 38% | F987[c] | Chronic kidney disease (mild-moderate) | Renal | 18% |
| F3203[a] | Anemia and acute kidney injury | Renal | 27% | F1170 | Anemia with immature granulocytes | Hematological | 18% |
| F3336 | Anemia with variable organ involvement | Hematological | 24% | F1070 | Mild anemia with normal chemistries | Hematological | 17% |
| F3656 | Hyperlipidemia, hypothyroidism screening | Metabolic | 24% | F3216 | Thrombocytopenia, abnormal RBC morphology | Hematological | 17% |
| F2804 | Post-op management after carotid stent | Cardiovascular | 23% | F864 | Biliary pathology leading to admission | GI | 17% |
| F1394 | Anemia evaluation, possible infection | Hematological | 21% | F2447 | Acute kidney injury with anemia | Renal | 17% |
| F1683 | Mild liver dysfunction with infection | GI | 20% | F3953[d] | Acute myocardial injury or infarction | Cardiovascular | 15% |
| F2279 | PSA measurement | Oncological | 19% | F3478[e] | Acidemia with anemia and coagulopathy | Multi-system | 14% |

