# OpenReview forum: "Probing Clinical Concepts in an EHR Foundation Model via Sparse Autoencoders"
_ICML.cc/2026/Workshop/FMSD — FMSD @ ICML 2026 Poster_

### Official Review · Reviewer_WCWG · 2026-05-15
**Interesting interpretability work on EHR foundation models, but results are not convincing**

**Rating:** 5
**Confidence:** 3

**Review:**

## Summary

This paper applies sparse autoencoders (SAEs) to the residual-stream activations of a transformer-based EHR foundation model trained on MIMIC-IV. The authors train SAEs on layers 1, 3, and 5, then interpret top-activating feature contexts using an LLM-based labeling pipeline. They argue that the resulting SAE features correspond to clinically meaningful concepts, including syndromes such as cardiorenal syndrome, DKA, SIRS, AMI, and DIC, and that cross-layer feature correlations plus activation patching provide evidence for feature-level information flow. The model is trained on next-code, next-value, and next-time prediction over structured EHR event streams with diagnoses, procedures, labs, and demographics.


## Strengths

The paper addresses an important problem- interpretability of structured EHR foundation models. Applying SAEs to EHR event-stream models is a promising direction, and the authors make a useful effort to adapt SAE-based interpretability to clinical data.

The qualitative examples are plausible. Many of the discovered features correspond to recurring clinical patterns one would expect in MIMIC-IV, such as anemia, kidney dysfunction, coagulopathy, infection workups, DKA-like metabolic emergencies, and cardiorenal/hepatorenal patterns.

The paper also goes beyond purely qualitative inspection by attempting different forms of validation, including ICD phenotype alignment, label-space clustering, cross-layer correlations, and activation patching. This is a good direction for making interpretability claims more testable.

## Weaknesses

The central claims appear stronger than the evidence supports. The mapping from SAE features to clinical concepts relies heavily on LLM interpretation of top-activating contexts. This is useful for hypothesis generation, but without clinician validation or robustness checks, it is hard to know whether the labels reflect true model features, dataset artifacts, or prompt-sensitive LLM summaries.

The external validation is weak. Only 90 out of 300 features are ICD-matchable, and the mean AUROC against ICD phenotypes is 0.541. This is better described as weak directional alignment rather than strong validation of the clinical semantics of the features. Intuitively, Im not sure why this should be so low.

The LLM labeling setup may also inflate interpretability. The LLM is shown windows containing events before and after the activation point, even though the model’s causal representation at that point only depends on prior context. This may help interpretation, but it makes the resulting label less clean as evidence for what the model had encoded at that event.

The activation patching experiments are interesting, but I am not fully convinced they establish clean clinical circuits. SAE decoder directions may be entangled, and ablating one direction may perturb the representation in ways that do not correspond to a clean intervention on a clinical concept. Stronger controls would be needed before making mechanistic claims.

A more fundamental concern is that structured EHR event streams are not directly analogous to natural language sequences. In language models, although interpretation is still difficult, the inputs are at least communicative tokens with relatively direct semantic content. In EHRs, however, event sequences reflect a mixture of patient physiology, clinician decision-making, ordering practices, documentation habits, care protocols, and site-specific workflows. As a result, SAE features labeled as clinical syndromes may instead correspond to recurring care trajectories or measurement/documentation patterns rather than pathophysiologic concepts. This weakens claims that the model has learned a clinical ontology or clinical mechanisms. I would encourage the authors to reframe the findings as candidate event-stream features or care-process patterns unless they can provide stronger validation against clinical definitions, clinician review, and multi-site replication.

Finally on a related note this study is limited to a single EHR dataset/site. Given the strong possibility of institution-specific ordering conventions, ICU workflows, and lab-ordering artifacts in MIMIC-IV, it is hard to separate general clinical concepts from site-specific patterns.


## Suggestions for Improvement

The paper would be substantially stronger with clinician validation of a subset of feature labels, pre-activation-only labeling experiments, SAE stability analysis across random seeds and dictionary sizes, and comparisons to simpler baselines such as PCA, clustering, or code co-occurrence. Replication on another EHR dataset would also help distinguish robust clinical concepts from MIMIC-specific artifacts.

The feature interpretation pipeline relies on a locally hosted Qwen 2.5 7B model, but the authors do not justify this choice or evaluate the reliability of the LLM labeler. Given that the central claims depend on clinical feature labels, comparison against stronger medical LLMs and/or clinician annotations would substantially strengthen the work.

An additional analysis I would love to see  is output-level activation patching- for features labeled as DKA, UTI, AMI, AKI, etc., ablate or amplify the SAE feature and measure whether the EHR-FM’s next-event predictions change in clinically specific directions. For example, ablating a DKA-labeled feature should reduce predicted probabilities of glucose/acidosis/insulin-related events more than unrelated events. This would provide more direct evidence that the feature has clinical and computational meaning, rather than only changing other SAE features. Im not fully convinced by the validation results presented

The authors should also soften claims around ‘monosemantic syndromes,’ ‘clinical ontology,’ and ‘syndrome discovery.’ The current evidence better supports the claim that SAEs surface plausible candidate clinical patterns, not that they definitively recover a validated clinical ontology.


## Overall Assessment

I lean reject with low confidence. The direction is promising and potentially useful for EHR foundation model interpretability, but the current validation is extremely thin and not strong enough to support the paper’s main claims. I see this as an interesting exploratory analysis rather than something clinically meaningful or generalizable.

---

### Official Review · Reviewer_aJEQ · 2026-05-20
**Solid SAE pipeline for an EHR transformer; external validity is thin**

**Rating:** 6
**Confidence:** 4

**Review:**

## Summary
Authors train a 27.5M-parameter, 6-layer causal transformer on MIMIC-IV in MEDS triplet format and fit TopK SAEs (k=32, dict 4096) on residual streams at layers 1, 3, 5. A local Qwen 2.5 7B labels 100 features per layer from top-activating patient contexts, and the resulting labels are validated three ways: held-out ICD AUROC, activation patching across layers, and a label-embedding UMAP. The main claims are that SAE features partition by activation prevalence rather than ICD hierarchy, that several named syndromes (cardiorenal, hepatorenal, SIRS, DKA, AMI, pancreatitis, etc.) surface as single monosemantic features with multi-finding syndromes concentrated deeper in the stack, and that high-correlation L1->L3->L5 chains carry interventional weight (69.7% of 400 source-target pairs show >5% downstream change).
## Strengths
- Sensible separation of concerns: pretrain, cache, fit SAEs offline, then label and validate.
- Three validation axes is more than most interpretability papers attempt at this scale.
- The random L1->L5 pair null in Appendix G (median r approx 0, 99th percentile r=0.56) gives a real baseline against which the labeled chains (r=0.58 to 0.95) clearly sit in the tail.
- Honest about limitations: single site, partial coverage, AUROC reported as modest rather than oversold.
- Reproduction code is there and looks complete: MEDS conversion, FM pretraining, SAE training, the LLM labelling driver, and the revision-pass analysis scripts.
## Areas for improvement
- The "EHR foundation model" framing is generous for a 27.5M-param, 6-layer model. Med-BERT, BEHRT and EHRMamba sit at much larger scale, and whether SAE findings transfer to those is the more interesting question. Some discussion of expected scaling would help.
- External validity is the weak link. AUROC 0.541 across 90 ICD-matchable features (79/90 above 0.5, p < 2 × 10^−5 one-sided binomial) is statistically nonzero but clinically thin. The paper attributes this to ICD coarseness, which is fair but the only thirdparty check on the LLM labels is then quite weak. Even a small clinician-adjudicated subset would have been more convincing.
- The 69.7% interventional headline overstates the underlying numbers. The median effect is "0.5 of 100 L5 targets shifted by >5%" per L1 ablation, which means most patches barely move the late stream. The story holds at the chain-by-chain level but the population claim sounds stronger than it is.
- Qwen 2.5 7B is a small LLM for clinical labeling. The prompt also primes a syndrome match (App. C asks for a "known clinical syndrome if applicable") which can anchor the model on named entities rather than describe what a feature actually fires on. A second labeler and a syndrome-free control prompt would tighten the discovery claim.
- The candidate cross-layer triples are pre-filtered to r>0.7 before patching, then patching results are tabulated. Selecting on the same statistic before testing inflates the apparent hit rate; the random-pair null in App. G mitigates this but does not remove it. Pre-registering the candidate set or holding out a chain set would be cleaner.
- Coverage is 100 of 4096 features per layer in the main pass (~2.4%). The repo contains n=500 scripts and the abstraction-tier chi-square in Figure 3 uses n=500, but the syndrome tables (Table 2, Table 3) appear to be from the n=100 pass. Please state explicitly which numbers come from which sample.
- The repo ships feature_eval/baselines.py with SHAP and attention-weight comparisons, but no baseline appears in the paper. A single comparison (raw residual probing or neuron-level monosemanticity on the same metric) would help motivate the SAE step rather than asking the reader to take it as given.
- Section 3.2 says SAEs train on the full 5M-vector pool without a held-out split. Downstream patient-cohort analyses are at patient level so they do generalise, but the alive/dead/L0 numbers in App. B are training-set figures.
## Detailed comments
- Figure 1 shows three hand-picked chains. How many candidate (L1, L3, L5) triples pass the r>0.7 filter, and what does the patching-effect distribution look like across all of them rather than the three exemplars?
- Table 2: the cardiorenal L5 feature fires on 42% of inputs. That activation rate sits more comfortably with a high-prevalence context detector than with a specific syndromic signal. Some discussion of how high firing rate squares with "single monosemantic feature for syndrome X" would help.
- Section 4: framing the result as EHR-FM learns an ontology distinct from ICD is fine, but an alternative reading is that features track lab/value patterns and admission contexts that span ICD chapters because real patients do. A baseline that clusters raw codes by patient co-occurrence would help distinguish "model learned something" from "ICD is anatomy-based".
- Please report base-model pretraining val loss and a downstream task number (e.g., mortality AUROC) so the reader can judge whether the FM under audit is a reasonable representative of the class.
- Minor: the citation to Modi et al. (medRxiv 2026.01.26.26344845) is real and relevant, worth a 1 sentence comparison since their work also uses SAEs on MIMIC-IV (clinical notes through MedGemma-27B, not event streams) which is the natural neighbour to the contribution here.
## Justification of score
Real work, fully reproducible, sensible methodology, and three validation axes. But the empirical case is thinner than the framing: external validity is weak, interventional effects are modest, coverage is partial, the labeling LLM is small and prompt-biased, and the audited FM is on the small end. For a non-archival workshop where modest contributions are explicitly welcome these are acceptable trade-offs, and the methodological contribution stands on its own. Marginally above threshold.

---

### Official Review · Reviewer_UqB8 · 2026-05-20
**Review for "Probing Clinical Concepts in an EHR Foundation Model via Sparse Autoencoders"**

**Rating:** 8
**Confidence:** 4

**Review:**

# Summary

This paper applies sparse autoencoders to an EHR foundation model trained on MIMIC-IV, extending interpretability to structured clinical event streams. The authors train TopK SAEs on residual-stream activations at three layers, and use an LLM pipeline to interpret features from top-activating patient contexts. Four findings reported: the model encodes named clinical syndromes as single monosemantic features distinct from ICD's hierarchy; features organized by activation prevalence rather than layer depth; cross-layer circuits compose lower-level signals into syndromes, validated via activation patching; and LLM labels align with held-out ICD phenotypes.

# Strengths

- This paper introduces a systematic application of SAE-based mechanistic interpretability to EHR foundation models, addressing a timely need given growing interest in clinical deployment.

- Claims are supported soundly along different complementary axes, including interventional, external, and internal consistency. Activation patching with random-pair null comparisons moves beyond correlational claims.

- There are clinically interesting findings, including cross-system syndromes as single features, with ontology implications.

# Areas for Improvement

- The external validity framing in the abstract ("feature activations align with held-out ICD phenotypes") is much stronger than the appendix description ("whether LLM-assigned labels have any systematic association with ICD-level phenotypes"). Aligning these would help readers calibrate what this axis tests.

- The related work critiques attention visualization, SHAP, and gradient attribution as inadequate, but the paper does not empirically compare SAE-derived features against these baselines.

- While the authors acknowledge in the Limitations that only a sample of features was interpreted, they do not specify the selection procedure for the 100 features per layer. This is distinct from the sample-size limitation the authors raise.

# Detailed Comments

- Regarding external validity test, it’s recommended to include the AUROC numbers and the appendix's narrower framing in the Results section.

- On the baseline comparison, an informative addition would be comparing TopK SAEs against simpler decomposition baselines, applied to the same cached residual-stream activations, using the same LLM interpretation pipeline.

- Add a single sentence in Section 3.3 stating the SAE feature selection procedure, and ideally report the distribution of the interpreted sample alongside that of the full dictionary. If feasible, a small follow-up experiment interpreting ~50 uniformly random features per layer would clarify whether the syndrome-discovery rate generalizes to the full dictionary.

# Justification of Score

Overall, this is a great work of systematic SAE-based interpretability analysis on EHR foundation models, which is a novel and timely contribution. Multi-axis validation seems rigorous. Clinically meaningful findings on cross system syndromes and prevalence-banded organization. Future improvements would benefit from baseline comparisons against other interpretability methods, and documentation of the feature selection protocol.